# Identification of illness representational patterns and examining differences of self-care behavior in the patterns in chronic kidney disease

Yuki Kajiwara [1,2]*, Michiko Morimoto[1]

**1** Faculty of Health Sciences, Okayama University, Okayama, Japan, **2** Graduate School of Health Sciences, Doctor's program, Okayama University, Okayama, Japan

* ykajiwara@okayama-u.ac.jp

**Data Availability Statement:** All relevant data are within the paper and its Supporting Information files.

## Abstract

Self-care behavior is considered important for preventing the progression of chronic kidney disease (CKD). Although lifestyle interventions are popular, they have not been sufficiently effective. According to studies on other chronic diseases, illness representation has been found to formulate a pattern, and self-care behavior could differ depending on the pattern, which suggests difference in self-care behavior based on illness representation. This study examined what kind of illness representational patterns exist among CKD patients and whether there is a difference in self-care behavior depending on the pattern. A survey was conducted from the beginning of June to the end of October 2019 on 274 CKD patients who were either outpatients or hospitalized at general hospitals in Western Japan. The Illness Perception Questionnaire-Revised was used to assess illness representation and the Japanese Chronic Kidney Disease Self-Care scale was used to assess self-care behavior. Two-stage cluster analysis was used to identify clusters. Cluster features were examined using analysis of variance and Tukey HSD tests. Differences in self-care behavior scores among identified clusters were investigated. Two hundred and forty-four questionnaires were received, and 212 were analyzed. Participants were aged 64.9±12.9, and the estimated glomerular filtration rate was 33.7±15.8. Three clusters were identified: Cluster 1 represented the difficulty of making sense of the changed condition caused by the disease and easily falling into misunderstanding; Cluster 2 represented patients with disease conditions that impacted their daily life and emotional responses; Cluster 3 represented the controllability and understandability of the disease. Total self-care behavior scores indicated a significant difference between Cluster 1 (52.1 ± 9.7) and Cluster 3 (57.7 ± 8.2). In conclusion, we showed that three representational patterns exist among CKD patients. In addition, a difference was found in self-care behavior depending on the illness representational pattern, suggesting the need to focus on illness representation.

**Funding:** This study was supported by a Grant-in-Aid For Scientific Research (C) from Japan Society for the Promotion of Science (JSPS KAKENHI Grant Number JP21K10712) awarded to Y.K. (https://www-shinsei.jsps.go.jp/kaken/index.html). No additional external funding was received for this study. The funder had no role in study design, data collection and analysis, decision to publish, or preparation of the manuscript.

**Competing interests:** The authors have declared that no competing interests exist.

## Introduction

Chronic kidney disease (CKD) is estimated to affect 850 million people, and the number of patients is increasing worldwide [1]. When CKD progresses, it leads to end-stage renal failure and requires renal replacement therapies (RRTs), such as dialysis and/or kidney transplantation. RRT not only places a psychological and economic burden on patients but also leads to increased expenses for medical systems [2]. Moreover, decreased glomerular filtration rate (GFR) is a risk factor for cardiovascular disease (CVD) and increases the risk of death in patients [2]. Reducing such risks and preventing and controlling the progression of CKD have become important issues worldwide.

In Japan, the estimated prevalence of CKD has increased from 13.3 million to 14.8 million over 10 years based on cohort studies [3]. The CKD Clinical Practice Guideline recommends these self-care behaviors, such as adherence to medication, diet management, exercise, smoking cessation, and blood pressure management [4]. However, CKD has poor subjective symptoms; continuing self-care behavior for it is difficult. Patients with end-stage renal failure are also reported to take treatment actions that are not appropriate [5]. To date, intervention studies on lifestyles have been conducted using self-care behaviors such as diet management and blood pressure management as outcomes for patients with CKD [6–10]; however, a meta-analysis has shown that the effects of this approach are limited, and educational interventions that focus only on lifestyle are insufficient to bring about behavioral change [11]. Under similar circumstances, in the case of diabetes and other chronic diseases, studies focusing on the patient's belief or perception of their disease have been advanced [12, 13], and the relationship with self-care behavior has also been examined [14]. Studies related to CKD patients' beliefs or perceptions of their disease have focused on the relationship between illness representation and the duration until the start of dialysis [15]; in addition, mortality is predicted [16] in CKD patients. In dialysis patients, a moderate relationship between the perception of illness and coping behavior has been reported [17]. When people suffer from a disease, they tend to generate specific patterns of belief, representation, and the perception of illness [16]. These beliefs and representations are conceptualized in the Common Sense Model (CSM), as shown by Leventhal et al. According to their model, illness representation guides coping behavior [18], and based on previous research [15–17], illness representation is inferred to be a factor that makes a difference in the self-care behavior of patients with CKD. In qualitative studies, patients were reported to find difficulty in understanding that CKD is a permanent disease; in understanding the severity of the disease because of its hard-to-discern symptoms, even if renal function is impaired; and in realizing the effects of maintaining renal function by changing their lifestyle [19, 20]. Muscat and her colleague focus on a pattern of illness representation called controllability, show that it is an important predictor of mortality risk, and suggest its association with self-care behavior [16]. However, this study also does not directly examine the relationship between CKD and self-care behavior. In patients with CKD, reports that directly examine the relationship between illness representation and self-care behavior are rare, and more studies are needed that will lead to changes in the self-care behavior of patients with CKD.

Illness representations construct multiple dimensions, such as how illness affects their lives (*consequence*), whether they can control the disease by themselves (*personal control*), and their emotional perception of it (*emotional representations*) [18, 21]. Previous studies in patients with other chronic diseases have examined the relationship between illness representation and self-care behavior in relation to each dimension, and a meta-analysis shows that this relationship is weak [22]. However, another meta-analysis has pointed out that since illness representation comprises multiple dimensions, extracting the two variables of each dimension and

outcome and examining the relationship between them is theoretically inconsistent [23]. Showing the relationship between two variables—one dimension and an outcome—can lead to misinterpretation [23]. In recent reviews, illness representation shows 2–3 patterns of characteristics based on cluster analysis, and differences in outcomes, such as quality of life and psychological burden, exist owing to differences in the characteristics of illness representation [24]. Although studies on outcomes of self-care behavior are limited, researchers have reported that patterns of illness representation are divided into three in diabetic patients and that differences exist in their self-care behavior [25]. Similarly, the pattern of illness representation in cardiac rehabilitation patients is divided into two, and differences in rehabilitation efforts exist depending on the pattern [26]. These reports indicate that different patterns of illness representation exist depending on the disease or on the patient's condition.

Therefore, in this study, we examine what kinds of representational patterns are present in patients with CKD before the start of dialysis. In addition, we examine whether differences exist in self-care behavior depending on the representational pattern. The representational patterns in CKD patients have not yet been shown. Moreover, differences in self-care behavior owing to representational patterns add a new perspective to existing interventions.

## Materials and methods

### Study design and participants

The study design followed a cross-sectional descriptive approach. The survey was conducted at Okayama university hospital located in West Japan. Eligible patients were on outpatient visits to, or were already in, the ward during the five-month period from June 19 to October 24, 2019 and met all the following selection criteria: clinically predicted to have a chronic course of the underlying disease leading to decreased renal function, having an estimated GFR (eGFR) of 60 mL/min/1.73 m$^2$, and being 20 years of age or older. The exclusion criteria were as follows: having started RRT; having difficulty answering the self-administered questionnaire owing to cognitive decline; undergoing treatment mainly for comorbidities; and mainly being treated with drug treatment, such as immunosuppressive drugs. All participants could read and write Japanese. Regarding the recruitment of participants, outpatients were recruited on the day of the outpatient consultation, and permission was obtained from the attending physician for the survey. After that, the survey was explained to outpatients. For inpatients, we confirmed whether there was a patient in charge who was hospitalized according to the outpatient day of the attending physician. After obtaining consent from the attending physician, the study was explained to the inpatients.

Sample sizes were calculated using Cohen's definition of effect size (f-test). A required sample size of 207 was calculated to produce a statistical power of 0.9 with a medium effect size of 0.25 and a significance level (p-value) of 0.05. In this study, 276 CKD patients were targeted, assuming a survey response rate of approximately 75%; however, the total final responses came to 274.

### Ethical considerations

This study complied with the Declaration of Helsinki and was approved by the Okayama University Clinical Research Review Committee (K1905-023). In the course of the survey, the researchers explained in writing and orally the purpose and significance of the research, method, viewing of information in medical records, and protection of personal information. The researcher distributed an anonymous self-administered questionnaire to those who expressed their intention to participate. Patients on outpatient visits were given one month to complete and return the questionnaire, while patients who were hospitalized were given 1–2

weeks in consideration of the burden on participants. Informed consent for this study was obtained verbally and in writing.

## Data collection

Participants' background factors included sex, age, body mass index (BMI), eGFR at the time of the survey, comorbidities, duration of disease, experience of receiving education on lifestyle for CKD from medical staff, living status (living together with families), and employment status. Information on BMI, eGFR at the time of the survey, and comorbidities was obtained from electronical medical records. Regarding the patient's perceived health status, we asked, "How was your overall health status over the past 1 month?" They were requested to respond on a 6-point scale, ranging from "extremely healthy" to "very poor."

The questionnaire was completed at home and mailed by the patient. Researchers collected information from electronic medical records for patients who checked the consent box on the questionnaire when they received it. For this reason, when distributing questionnaires, it was necessary to assign IDs to patients and match them with examination data on electronic medical records. After matching information such as test values from consenting patients, the researchers erased all patient-identifying information. Thus, researchers had access to patient-identifiable information only when gathering information from electronic medical records.

## Data measurements

**Illness representation.** The Illness Perception Questionnaire-Revised (IPQ-R) was used to evaluate illness representation [21]. The IPQ-R has been used to assess patients' representation in multiple chronic diseases, including CKD, and is a valid scale for measuring illness representation in CKD patients. In addition, it has been translated into Japanese by Katayama et al., and its content validity and reliability have been confirmed using a retesting method [27]. The IPQ-R is composed of three sections. The first section is "*identity*," for which a "yes/no" binary is used for each item; the score indicates the number of symptoms associated with the disease. The second section is the assessment of cognitive condition in each dimension (*timeline acute/chronic*, *consequence*, *personal control*, *treatment control*, *illness coherence*, *timeline cyclical*, *emotional representations*). Question items include, for example, "My kidney disease has serious consequences for my life" (*consequence*) and "I can decide whether what I do improves or worsens my kidney disease" (*personal control*). The responses to the items are on a 5-point scale, ranging from "strongly disagree" to "strongly agree" [21, 28]. The total score is calculated for each dimension; what the score indicates is different for each dimension. *Timeline acute/chronic* represents the belief regarding whether the disease follows a chronic course, "*consequence*" represents the belief that the disease has serious consequences on one's life, and "*personal control*" represents the belief about control over the disease. Additionally, "*treatment control*" represents the belief that treatment is effective in controlling the disease; "*illness coherence*" represents the patient's beliefs regarding making sense of their condition caused by the disease; "*timeline cyclical*" represents the belief that medical conditions and symptoms are not constant; and "*emotional representations*" indicate negative emotional reactions to illness. The third dimension is "*cause*," which indicates the cause of the disease recognized by the patient. However, in CKD, the causes differ depending on the underlying disease or when multiple causes are mixed. Therefore, we did not use "*cause*" in this study.

**Self-care behavior.** To assess patients' self-care behavior, we used the Japanese version of the Chronic Kidney Disease Self-Care Scale (CKDSC-J). The reliability and validity of the original scale (i.e., CKDSC), developed by Wang et al. [29], were confirmed. In addition, when the CKDSC-J was developed, its structural validity was confirmed by assuming a second-order

factor model; the one-dimensionality of the scale was also confirmed [30]. The CKDSC-J consists of 15 items and five factors: "medication control," "diet," "exercise," "smoking cessation," and "self-monitoring of blood pressure." Examples of question items include: "I follow the restricted diet rules for kidney disease, such as protein restriction, when eating meals" and "I measure (check) my blood pressure myself." Each item is evaluated on a 5-point scale from "never" to "always." The higher the score on the CKDSC-J, the more the patient is engaged in self-care behavior.

## Data analysis

For the analysis of data, descriptive statistics, cluster analysis, one-way analysis of variance (ANOVA), and Tukey Honestly Significant Difference (HSD) multiple comparison test were performed using SPSS ver. 26.0. We calculated the mean and standard deviation (SD) for each dimension of the background factor and IPQ-R and performed descriptive statistics. To categorize disease cognition, we performed a "two-stage cluster analysis," with reference to previous studies [31]. First, we performed hierarchical cluster analysis using Ward's method and examined the number of clusters. Next, we performed a non-hierarchical cluster analysis using the K-means method, extracted patterns of illness representation, confirmed that the population was roughly the same as the result of the hierarchical cluster, and determined the pattern of illness representation. To understand the characteristics of each extracted pattern, one-way ANOVA was used to examine the differences between groups in each of the eight dimensions of illness representation, followed by Tukey HSD multiple comparison test. In addition, since illness representation has dimensions with different scores, we calculated the Z-score and examined the characteristics of each pattern together with the results of one-way ANOVA. To examine whether a difference existed in self-care behavior depending on illness representation, the self-care behavior score for each representational pattern extracted was examined along with the one-way ANOVA analysis, followed by Tukey HSD multiple comparison test. The statistical significance level was 5%, and the test was two-sided.

## Results

A total of 244 people responded to the questionnaire (89.1% response rate). Of these, questionnaires for 212 participants with valid and complete responses on IPQ-R or CKDSC-J were analyzed (valid response rate 77.4%). Table 1 shows the characteristics of the participants. Their age was 64.9 ± 12.9 years, and the eGFR was 33.7 ± 15.8 mL/min/1.73m$^2$ at the time of the survey; 36.3% of them had been diagnosed with CKD for 1 to 5 years. Moreover, 92.0% had an experience of receiving education on lifestyle for CKD from medical staff. More than 70% of the respondents answered that their health status in the past 1 month was "healthy," "very healthy," or "extremely healthy."

### Difference in illness representational patterns and their features in patients with CKD

Three clusters—Cluster 1 (n = 66), Cluster 2 (n = 63), and Cluster 3 (n = 83)—were extracted by two-stage cluster analysis. Between-group differences existed in all dimensions of illness representation among the three clusters (*identity*: F = 31.51, p < .001; *timeline acute/chronic*: F = 51.42, p < .001; *consequence*: F = 79.22, p < .001, *personal control*: F = 43.37, p < .001, *treatment control*: F = 33.66, p < .001, *illness coherence*: F = 40.18, p < .001, *timeline cyclical*: F = 4.45, p = .013, *emotional representations*: F = 57.51, p < .001). The one-way ANOVA and the Tukey HSD multiple comparison test revealed that Cluster 1 had the lowest scores among the three clusters for "*timeline acute/chronic*," "*personal control*," and "*illness coherence*;"

**Table 1. Participant characteristics (N = 212).**

| Variable | | N | | % |
|---|---|---|---|---|
| **Sex** | Male | 140 | | 66.0 |
| | Female | 72 | | 34.0 |
| **Age (years), M ± SD** | | 64.9 | ± | 12.9 |
| **BMI (kg/m$^2$), M ± SD** | | 23.5 | ± | 4.1 |
| **eGFR (mL/min/1.73 m$^2$), M ± SD, N = 211[a]** | | 33.7 | ± | 15.8 |
| **CKD stage, N = 211 [a, b]** | G3a | 62 | | 29.4 |
| | G3b | 62 | | 29.4 |
| | G4 | 55 | | 26.0 |
| | G5 | 32 | | 15.2 |
| **Comorbidity[c]** | Hypertension | 167 | | 78.8 |
| | Hyperlipidemia | 111 | | 52.4 |
| | Hyperuricemia | 100 | | 47.2 |
| | Diabetes | 66 | | 31.1 |
| | Heart disease | 43 | | 20.3 |
| **CKD duration (years)** | Less than 1 year | 23 | | 10.8 |
| | 1–5 | 77 | | 36.3 |
| | 5–10 | 43 | | 20.3 |
| | 10–15 | 20 | | 9.4 |
| | 15–20 | 12 | | 5.7 |
| | 20 or more | 37 | | 17.5 |
| **Educating patients about CKD[d]** | | 195 | | 92.0 |
| **Educator[e]** | Physician | 173 | | 81.6 |
| | Nurse | 43 | | 20.3 |
| | Nutritionist | 122 | | 57.5 |
| **Self-rated health status[f]** | Extremely healthy | 3 | | 1.4 |
| | Very healthy | 30 | | 14.2 |
| | Healthy | 119 | | 56.1 |
| | Little poor | 52 | | 24.5 |
| | Poor | 8 | | 3.8 |
| | Very poor | 0 | | |
| **Living status, N = 210** | Living alone | 33 | | 15.7 |
| | Living with families | 177 | | 84.3 |
| **Employment status, N = 207** | Employed | 104 | | 50.3 |
| | Unemployed | 54 | | 26.1 |
| | Retired | 20 | | 9.6 |
| | Other | 29 | | 14.0 |

BMI, body mass index; CKD, chronic kidney disease; eGFR, estimated glomerular filtration rate; M, mean; SD, standard deviation.

[a] Mean values were calculated from and CKD stage in patients with available eGFR values measured at the time of the survey.

[b] CKD stage was categorized by eGFR values based on the Kidney Disease Improving Global Outcomes (KDIGO) clinical practice guidelines (45–59 [G3a], 30–44 [G3b], 15–29 [G4], <15 [G5] ml/min/1.73 m$^2$). Patients who met the criteria at recruitment but had an eGFR $\geq$ 60 mL/min/1.73 m$^2$ in the survey were included in G3a (N = 7).

[c] Comorbidities indicate multiple co-occurring diseases per patient in the electronic medical record.

[d] Experience of receiving education on lifestyle for CKD from medical staff.

[e] Occupations of medical staff who educated patients.

[f] Patients evaluated their health status for the past 1 month.

**Table 2. Comparison of illness perception among clusters.**

| Total score for each IPQ-R dimension | Total N = 212 | | Cluster 1[a] N = 66 | | Cluster 2[b] N = 63 | | Cluster 3[c] N = 83 | | F (p-value) |
|---|---|---|---|---|---|---|---|---|---|
| | M | (SD) | M | (SD) | M | (SD) | M | (SD) | |
| **Identity** | 1.8 | (2.6) | 1.1 | (1.6) | 3.7 | (3.4) | 1.0 | (1.6) | 31.51*** (< .001) |
| | | | | | | | | | a, c < b |
| **Timeline acute/chronic** | 25.0 | (4.3) | 21.8 | (3.7) | 28.2 | (2.3) | 25.2 | (4.2) | 51.42*** (< .001) |
| | | | | | | | | | a < b, c; c < b |
| **Consequences** | 17.7 | (4.8) | 15.4 | (3.5) | 22.6 | (3.6) | 15.8 | (3.8) | 79.22*** (< .001) |
| | | | | | | | | | a, c < b |
| **Personal control** | 21.4 | (3.5) | 18.7 | (2.6) | 21.8 | (3.6) | 23.2 | (2.7) | 43.37*** (< .001) |
| | | | | | | | | | a < b, c; b < c |
| **Treatment control** | 18.1 | (3.1) | 17.0 | (2.8) | 16.8 | (3.3) | 20.0 | (2.2) | 33.66*** (< .001) |
| | | | | | | | | | a, b < c |
| **Illness coherence** | 17.5 | (2.7) | 15.6 | (2.1) | 17.3 | (2.6) | 19.1 | (2.3) | 40.18*** (< .001) |
| | | | | | | | | | a < b, c; b < c |
| **Timeline cyclical** | 9.3 | (2.6) | 9.8 | (2.3) | 9.6 | (2.7) | 8.7 | (2.5) | 4.45* (= .013) |
| | | | | | | | | | c < a |
| **Emotional representations** | 17.4 | (5.2) | 17.1 | (4.4) | 21.8 | (4.3) | 14.3 | (3.9) | 57.51*** (< .001) |
| | | | | | | | | | a, c < b; c < a |

IPQ-R, Illness Perception Questionnaire-Revised; M, mean; SD, standard deviations.

Statistically significant differences between clusters were found using the Tukey–Kramer test.

p-values were calculated based on ANOVA with the Tukey–Kramer test.

*Significant difference (p < .05)

***Significant difference (p < .001).

Cluster 2 had the highest scores among the three clusters for "*identity*," "*timeline acute/chronic*," "*consequence*," and "*emotional representations*;" and, interestingly, Cluster 3 had higher scores for "*personal control*," "*treatment control*," and "*illness coherence*" than the other two clusters (Table 2). The Z-score results are shown in Fig 1. Based on these results, Cluster 1 represents the difficulty of making sense of the changed condition caused by the disease and easily falling into misunderstanding, Cluster 2 represents patients with disease conditions that have impacted their daily life and emotional responses, and Cluster 3 represents the controllability and understandability of the disease.

### Difference in self-care behavior scores among representational patterns

The one-way ANOVA analysis, followed by Tukey HSD multiple comparison test, showed that the self-care behavior score was higher in Cluster 3 than in Cluster 1 (p < .001). No statistically significant differences were found between Cluster 2 and the other clusters. The self-care behavior scores of Clusters 1, 2, and 3 were 52.1 (SD = 9.7), 54.6 (SD = 8.6), and 57.7 (SD = 8.2), respectively (Table 3).

### Discussion

First, this study identified three distinct illness representational patterns among CKD patients: Cluster 2 with a high identity, timeline acute/chronic, consequence, and emotional representation; Cluster 3 with a high personal control, treatment control, and illness coherence; and Cluster 1 with a low timeline acute/chronic, personal control, and illness coherence. Similarly, a previous review using the cluster approach with IPQ-R identified two or three patterns [24].

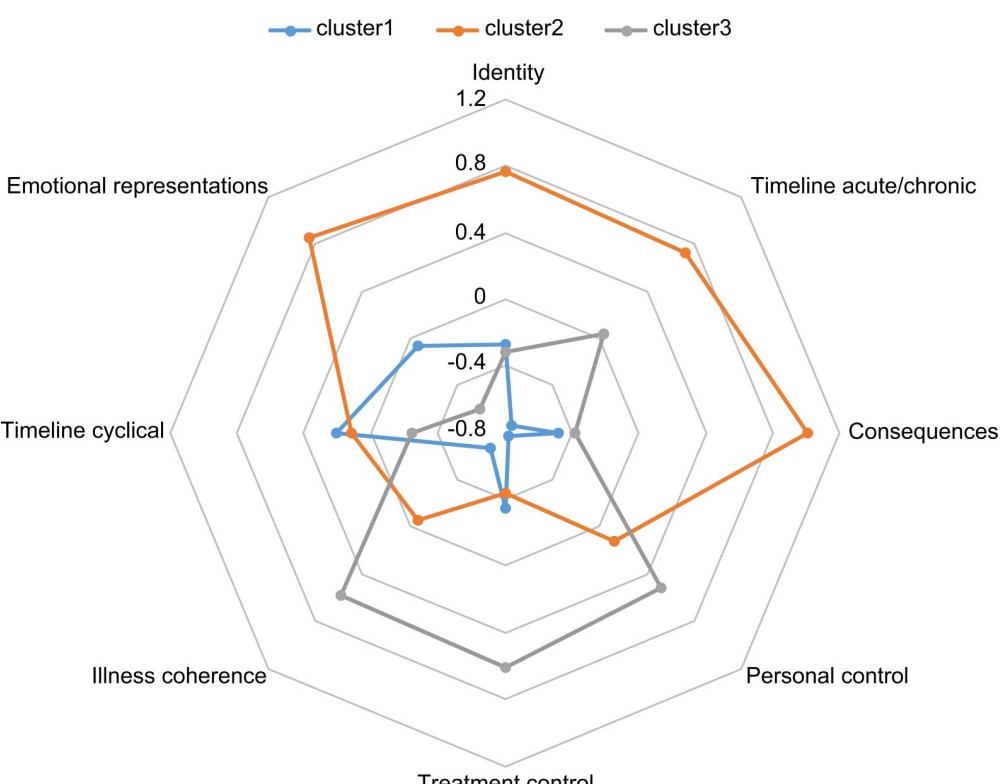

**Fig 1. Comparison of each domain of the IPQ-R mean scores (z-scores) by clusters.** Cluster 1 represented the difficulty of making sense of the changed condition caused by the disease and easily falling into misunderstanding (blue). Cluster 2 represents patients with disease conditions that have impacted their daily life and emotional responses (orange). Cluster 3 represents the controllability and understanding of the disease (gray). IPQ-R, Illness Perception Questionnaire-Revised.

Regarding the feature of Cluster 2, Hagger et al. reported that people who were higher on the domains of identity, timeline acute/chronic, consequence, and emotional representations in Cluster 2 (1) had representation of health threats regarding their disease and (2) tended to strongly recognize health threats and deal with them [23, 32]. In contrast, Cluster 3 indicated higher personal control, treatment control, and illness coherence among the three patterns, representing patients who have control over their disease. Similarly, people who have the representation of controllability tend to have trouble in recognizing health threats (identity, timeline acute/chronic, emotional representation) [23, 32]. In our study, the dimensions related to health threats were higher in Cluster 2 than in Cluster 3, showing trends similar to

**Table 3. Comparison of CKDSC-J score by cluster.**

|  | Total N = 212 | | Cluster 1[a] N = 66 | | Cluster 2[b] N = 63 | | Cluster 3[c] N = 83 | | F (p-value) |
|---|---|---|---|---|---|---|---|---|---|
| Outcome variables | M | (SD) | M | (SD) | M | (SD) | M | (SD) | |
| CKDSC-J score | 55.0 | (9.1) | 52.1 | (9.7) | 54.6 | (8.6) | 57.7 | (8.2) | 7.31*** (< .001) |
|  | | | | | | | | | a < c |

CKDSC-J, Chronic Kidney Disease Self-Care Scale-Japanese version; M, mean; SD, standard deviation.

Statistically significant differences between clusters were found by the Tukey–Kramer test.

p-values were calculated based on ANOVA with the Tukey–Kramer test.

***Significant difference (p < .001).

those of previous reports. In addition, previous reports that examined representational patterns in other chronic diseases, such as diabetes [25], myositis [33], and Chronic Obstructive Pulmonary Disease [34], indicated similar representational patterns of health threats and controllability. As patients with chronic diseases live with their disease, they are affected physically, psychologically, and socially by changes in their condition [35–37] but can engage in self-care behavior to deal with these changes and steady their life. While some patients with a chronic condition have a representation of health threats by changing their disease condition, others with the representation of controllability control their disease by engaging in self-care behavior to steady their life. For this, indicating that the representational patterns of Clusters 2 and 3 may also reflect the nature of a chronic disease.

Cluster 1 had the lowest score among the three patterns in terms of timeline acute/chronic, personal control, and illness coherence, implying that those following the pattern of Cluster 1 have poor representation of the chronic course of their disease, difficulty in controlling the disease on their own, and little ability to make sense of their condition caused by the disease. CKD is a disease with few subjective symptoms until it becomes severe [4]. In qualitative studies, patients report not knowing whether they are truly ill because they do not feel any impact on their lives; furthermore, as renal function declines without symptoms, they tend to imagine that the disease is not serious, and they consider their health to be good [19, 20]. Though CKD generally follows a chronic course, and its progression can be delayed through the individual's self-care behavior, participants in Cluster 1 showed a poor belief in the chronic progression of CKD and in the ability to control the disease themselves. This suggests that the patients in Cluster 1 had trouble making sense of their changed condition caused by the disease (e.g., changes in their physical, psychological, and situational conditions), as well as understanding their feelings in the context of the changed conditions; thus, misunderstandings were highly likely to occur.

In this study, a difference existed in the total scores for self-care behavior between Clusters 1 and 3, which showed differences in personal control, treatment control, and illness coherence, suggesting that differences in controllability brought about differences in self-care behavior. According to a previous report, illness representational patterns with controllability focus on the problem and aim to solve it [38]. In the case of CKD, self-care behaviors can be the solution to the problem of CKD progression. Based on this, we suggest that the difference in self-care behavior is caused by a difference between Clusters 3 and 1, which indicates difference of strength in the illness representational pattern of controllability, while the total score of self-care behavior in Cluster 2 shows no difference when compared to Clusters 1 and 3. In promoting self-care behaviors, multiple theories have assumed that these behaviors are caused by heightening belief in the health threat or by the patient's perceived seriousness about their disease, including the Health Belief Model. However, studies examining the approach of behavioral change have reported that interventions that increase only patient's perceptions of the health threat, or the seriousness of their disease, do not affect behavioral change sufficiently [39]. This study also showed that the pattern shown by Cluster 3 participants, indicates a strong belief in control over the disease. This result suggests that a belief in control over illness is important in enhancing self-care behavior.

In this study, we showed the possibility that the illness representation of CKD patients can be understood as three patterns. The relationship between illness representation and self-care behavior has not been shown directly yet. Though this is also the case in this study, we indicated a new finding that representational patterns in CKD can be observed and that self-care behavior differs depending on the pattern. In CKD patients, diverse beliefs have been reported in qualitative studies [19, 20, 40]. Evaluating the characteristics of patients' beliefs, through three illness representational patterns, will lead to a deeper understanding of the patients' beliefs for medical staff.

To prevent progression and aggravation of CKD, encouraging self-care behavior in patients is critical. However, educational intervention that focuses only on lifestyle is considered insufficient to bring about behavioral change. Under these circumstances, the fact that self-care behavior differs depending on the characteristics of the patient's belief suggests the possibility that illness representation can be a factor regulating self-care behavior in CKD patients. This indicates the need to focus on illness representation in promoting self-care behavior in CKD patients. In addition, the knowledge provided by our study can contribute to examining interventions for illness representation, as a new perspective.

This study had a few limitations. Specifically, the results of this study are expected to help medical staff understand and support CKD patients with decreased renal function; however, this study was conducted at only a single general hospital where many specialists in nephrology provide outpatient care. In addition, more than 90% of patients in this study had experience of receiving education on lifestyle for CKD from medical staff, and this may have impacted their beliefs or their self-care behavior. Furthermore, this was a cross-sectional study. In the future, verifying whether self-care behavior differs depending on the pattern of illness representation will be necessary, using a longitudinal design.

## Conclusion

In this study, we showed that three patterns of beliefs regarding CKD can be observed. In particular, Cluster 3 was identified as having a high degree of controllability, which is considered important for patients to engage in self-care behaviors in order to prevent progression of CKD. As a difference existed in the self-care behavior score by the characteristics of illness representation, we suggest that focusing on illness representation to promote self-care behavior by patients is necessary.

## Supporting information

**S1 Checklist. STROBE statement—checklist of items that should be included in reports of observational studies.**
(DOCX)

**S1 Dataset.**
(XLSX)

## Acknowledgments

The author would like to thank the participating patients for answering the questionnaire and the staff of the Okayama university hospital, especially Dr. Jun Wada and Dr. Kathuyuki Tanabe.

## Author Contributions

**Data curation:** Yuki Kajiwara, Michiko Morimoto.

**Formal analysis:** Yuki Kajiwara, Michiko Morimoto.

**Funding acquisition:** Yuki Kajiwara, Michiko Morimoto.

**Investigation:** Yuki Kajiwara, Michiko Morimoto.

**Methodology:** Yuki Kajiwara, Michiko Morimoto.

**Project administration:** Yuki Kajiwara.

**Writing – original draft:** Yuki Kajiwara.

**Writing – review & editing:** Yuki Kajiwara, Michiko Morimoto.

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
