## [Decision Letter · Decision Letter 0]

28 Dec 2022

PONE-D-22-27594Identification of illness representational patterns and examining differences of self-care behavior in the patterns in chronic kidney diseasePLOS ONE

Dear Dr. Kajiwara,

Thank you for submitting your manuscript to PLOS ONE. After careful consideration, we feel that it has merit but does not fully meet PLOS ONE’s publication criteria as it currently stands. Therefore, we invite you to submit a revised version of the manuscript that addresses the points raised during the review process.

We look forward to receiving your revised manuscript.

Kind regards,

Yuri Battaglia

Academic Editor

PLOS ONE

Journal Requirements:

Reviewers' comments:

Reviewer's Responses to Questions

**Comments to the Author**

1. Is the manuscript technically sound, and do the data support the conclusions?

Reviewer #1: Yes

Reviewer #2: Yes

2. Has the statistical analysis been performed appropriately and rigorously? 

Reviewer #1: Yes

Reviewer #2: Yes

3. Have the authors made all data underlying the findings in their manuscript fully available?

Reviewer #1: Yes

Reviewer #2: No

4. Is the manuscript presented in an intelligible fashion and written in standard English?

Reviewer #1: Yes

Reviewer #2: Yes

5. Review Comments to the Author

Reviewer #1: Kajiwara and Morimoto carried out a cross-sectional study examining the representational patterns present in patients with non-dialysis CKD and related self-care behavior. In a cohort of 212 patients, they identified three representational patterns with significant difference in self-care behavior related to these patterns.

This manuscript is really interesting, but I have some concerns. Briefly, my comments:

1) The Introduction and Discussion Section are not so readable because too long. I suggest to the Authors to be more concise.

2) CKD is a disease with a broad spectrum. Is it possible to evaluate if there are any differences in representational patterns in relationship with CKD stage or CKD cause?

Reviewer #2: Thanks for your excellent work in which you examined what kind of representational patterns are present in patients with chronic kidney disease and the differences in self-care behavior according to them.

I wonder if there are any changes in representational patterns according to socio-demographic variables (e.g. less controllability in patients living alone vs living with family) or clinical variables (e.g. does illness representation change according to CKD duration?).

I think it would be interesting in the future to explore the relationship between illness representation and self-care behavior with other analyses, like network analyses.

6. PLOS authors have the option to publish the peer review history of their article (what does this mean?). If published, this will include your full peer review and any attached files.

Reviewer #1: No

Reviewer #2: **Yes: **Luigi Zerbinati

---

## [Author Response · Author response to Decision Letter 0]

27 Feb 2023

Response to the reviewer comments is attached herewith as a separate document.

---

## [Editor Report · Decision Letter 1]

14 Mar 2023

Identification of illness representational patterns and examining differences of self-care behavior in the patterns in chronic kidney disease

PONE-D-22-27594R1

Dear Dr. Kajiwara,

We’re pleased to inform you that your manuscript has been judged scientifically suitable for publication and will be formally accepted for publication once it meets all outstanding technical requirements.

Kind regards,

Yuri Battaglia

Academic Editor

PLOS ONE

---

## [Editor Report · Acceptance letter]

20 Mar 2023

PONE-D-22-27594R1 

Identification of illness representational patterns and examining differences of self-care behavior in the patterns in chronic kidney disease 

Dear Dr. Kajiwara:

I'm pleased to inform you that your manuscript has been deemed suitable for publication in PLOS ONE. Congratulations! Your manuscript is now with our production department. 

Kind regards, 

on behalf of

Prof. Yuri Battaglia 

Academic Editor

PLOS ONE